# Diversity of Late Blight Resistance Genes in the VIR Potato Collection

**DOI:** 10.3390/plants12020273

**Published:** 2023-01-06

**Authors:** Elena V. Rogozina, Alyona A. Gurina, Nadezhda A. Chalaya, Nadezhda M. Zoteyeva, Mariya A. Kuznetsova, Mariya P. Beketova, Oksana A. Muratova, Ekaterina A. Sokolova, Polina E. Drobyazina, Emil E. Khavkin

**Affiliations:** 1N.I. Vavilov Institute of Plant Genetic Resources (VIR), St. Petersburg 190000, Russia; 2Institute of Phytopathology, Bol’shiye Vyazemy, Moscow 143050, Russia; 3Institute of Agricultural Biotechnology, Moscow 127550, Russia

**Keywords:** *Phytophthora infestans*, *Solanum* section *Petota*, genetic diversity, wild potato relatives, germplasm enrichment, late blight resistance, resistance genes

## Abstract

Late blight (LB) caused by the oomycete *Phytophthora infestans* (Mont.) de Bary is the greatest threat to potato production worldwide. Current potato breeding for LB resistance heavily depends on the introduction of new genes for resistance to *P. infestans* (*Rpi* genes). Such genes have been discovered in highly diverse wild, primitive, and cultivated species of tuber-bearing potatoes (*Solanum* L. section *Petota* Dumort.) and introgressed into the elite potato cultivars by hybridization and transgenic complementation. Unfortunately, even the most resistant potato varieties have been overcome by LB due to the arrival of new pathogen strains and their rapid evolution. Therefore, novel sources for germplasm enhancement comprising the broad-spectrum *Rpi* genes are in high demand with breeders who aim to provide durable LB resistance. The Genbank of the N.I. Vavilov Institute of Plant Genetic Resources (VIR) in St. Petersburg harbors one of the world’s largest collections of potato and potato relatives. In this study, LB resistance was evaluated in a core selection representing 20 species of seven *Petota* series according to the Hawkes (1990) classification: *Bulbocastana* (Rydb.) Hawkes, *Demissa* Buk., *Longipedicellata* Buk., *Maglia* Bitt., *Pinnatisecta* (Rydb.) Hawkes, *Tuberosa* (Rydb.) Hawkes (wild and cultivated species), and *Yungasensa* Corr. LB resistance was assessed in 96 accessions representing 18 species in the laboratory test with detached leaves using a highly virulent and aggressive isolate of *P. infestans*. The *Petota* species notably differed in their LB resistance: *S. bulbocastanum* Dun., *S. demissum* Lindl., *S. cardiophyllum* Lindl., and *S. berthaultii* Hawkes stood out at a high frequency of resistant accessions (7–9 points on a 9-point scale). Well-established specific SCAR markers of ten *Rpi* genes—*Rpi-R1*, *Rpi-R2/Rpi-blb3*, *Rpi-R3a*, *Rpi-R3b*, *Rpi-R8*, *Rpi-blb1/Rpi-sto1*, *Rpi-blb2,* and *Rpi-vnt1*—were used to mine 117 accessions representing 20 species from seven *Petota* series. In particular, our evidence confirmed the diverse *Rpi* gene location in two American continents. The structural homologs of the *Rpi-R2*, *Rpi-R3a*, *Rpi-R3b,* and *Rpi-R8* genes were found in the North American species other than *S. demissum*, the species that was the original source of these genes for early potato breeding, and in some cases, in the South American *Tuberosa* species. The *Rpi-blb1/Rpi-sto1* orthologs from *S. bulbocastanum* and *S. stoloniferum* Schlechtd et Bché were restricted to genome B in the Mesoamerican series *Bulbocastana*, *Pinnatisecta,* and *Longipedicellata*. The structural homologs of the *Rpi-vnt1* gene that were initially identified in the South American species *S. venturii* Hawkes and Hjert. were reported, for the first time, in the North American series of *Petota* species.

## 1. Introduction

The potato belongs to major staple crops that sustainably feed the world [1,2]. To attain this goal, current potato breeding heavily relies on the incessant inflow of new genes that stand for durable resistance to abiotic and biotic adverse factors and higher tuber yield of better quality.

In Russia, with its annual potato production of about 22 million metric tons and consumption of 110 kg per capita per year (https://www.potatopro.com/russian-federation/potato-statistics; assessed on 4 January 2023), this crop is the mainstay of food security. Here, as well as worldwide, late blight (LB) caused by the oomycete *Phytophthora infestans* (Mont.) de Bary is a major threat to potato crops, and the dominant race-specific genes for resistance to *P. infestans* (*Rpi* genes) have been most vigorously searched for by geneticists and explored by breeders [3,4,5,6,7,8,9]. These genes are found in wild, primitive, and cultivated tuber-bearing species of *Solanum* L. (section *Petota* Dumort.), and many of these genes have been successfully introgressed into potato cultivars by sexual and somatic hybridization or by genetic transformation. By now, over 20 *Rpi* genes, all belonging to the CC-NB-LRR group, have been established and investigated in great detail in diverse *Petota* species, particularly those belonging to the North American and Mesoamerican series *Bulbocastana* (Rydb.) Hawkes, *Demissa* Buk. and *Longipedicellata* Buk. and to the South American series *Tuberosa* (Rydb.) Hawkes [2,5,6,9,10,11,12,13,14]. Outside two American continents and beyond the section *Petota*, we find the homologous *Rpi* genes in such widespread species as *S. americanum* Mill. and *S. nigrum* L. [15,16].

Novel sources of LB resistance for germplasm enhancement are in high demand with breeders because of pathogen migration and rapid evolution; therefore, preferred are potato species and *Rpi* genes that have not been previously involved in large-scale breeding. With the advent of the “omics” era, the molecular genetic approach to broadening the basic germplasm resource for potato breeding is speedily gaining momentum. Especially significant in this context is new knowledge on the structure and function of the *Rpi* genes; no less important are the advanced technologies of mining for these genes in germplasm collections and introducing the beneficial genes to practical breeding. As a result, even although most *Petota* species have not been sufficiently researched, we evidence the expanding list of prospective breeding sources, with many promising accessions and particular individuals further maintained as clones that contribute new valuable alleles for germplasm enrichment [2,10,17,18,19,20,21,22,23,24,25].

Several World Genbanks in Europe and America maintain vast genetic collections of tuber-bearing *Solanum* species, and numerous accessions of these species have been screened for LB resistance with specific and well-characterized pathogen isolates. The wild potato germplasm maintained in the local collections seems to present another exciting treasure trove of the *Rpi* genes. As a result, many resistant accessions have been earmarked across the *Petota* species [25,26,27,28,29,30,31].

To mine for the *Rpi* genes, researchers utilize such specific technologies as PCR amplification, e.g., with the sequence characterized amplified region (SCAR) markers [3,22,32,33,34], and the resistance gene enrichment sequencing technologies [7,15,35,36]. An alternative approach, effectoromics, employs avirulence (*Avr*) genes of *P. infestans* matching the corresponding *Rpi* genes [5] as a tool for recognizing new *Rpi* genes and, what is most significant, discerning between the functional genes and their structural homologs, which may possess yet unknown functions. The outcome of such screening is most vividly exemplified in the cases of the gene cluster on chromosome 4 comprising the *Rpi-R2* gene and its numerous orthologs in the Mexican species *S. bulbocastanum* Dun., *S. demissum* Lindl., *S. edinense* Berth., *S. hjertingii* Hawkes, and *S. schenckii* Bitter [12,37,38], the orthologous *Rpi-blb1/Rpi-sto1* genes on chromosome 8 in the Mexican species *S. bulbocastanum*, *S. stoloniferum* Schlechtd. et Bché, *S. papita* Rydb. and *S. polytrichon* Rydb. [3,5,38,39], and several *Rpi* genes in the cluster on the distal part of chromosome 9, including *Rpi-R8* and *Rpi-R9a* from *S. demissum*, *Rpi-vnt1* from *S. venturii* Hawkes and Hjert. and *Rpi-ver1* from *S. verrucosum* Schlechtd [40,41,42,43,44].

The VIR Genbank is one of the five largest ex situ collections of potato and related tuber-bearing species, with its oldest accessions traced back to 1926–1927 [45]. It is one of the world’s best collections of wild, primitive, and cultivated species, which have been widely deployed in germplasm improvement. In this collection, the North American potato species are classified according to Bukasov [46], and the South American wild species according to Gorbatenko [47]. However, for the convenience of the foreign reader, in this communication, we list the names of species and their series positions in accordance with the system by Hawkes [48].

The goal of our study was a first reconnaissance of the VIR potato collection in order to select the most attractive genotypes for further detailed research. For such selection, we combined two independent strategies: the phenotypical evaluation of late blight resistance in the detached leaf test with a highly virulent and already sufficiently investigated *P. infestans* isolate—and the molecular screening with well-validated SCAR markers. Both technologies are open to many criticisms; nevertheless, when combined, such data are very practical for the wide-range screening of new plant germplasm.

Here we report the evidence from our study of LB resistance in the core selection from the VIR collection of wild potatoes (Appendix A). The study covered 20 species of seven *Petota* series according to the classification by Hawkes [48]: *Bulbocastana* (Rydb.) Hawkes, *Demissa* Buk., *Longipedicellata* Buk., *Maglia* Bitt., *Pinnatisecta* (Rydb.) Hawkes, *Tuberosa* (Rydb.) Hawkes (wild and cultivated species), and *Yungasensa* Corr., with a total of 201 accessions (plant IDs are assigned to the collecting numbers). Each plant ID is represented by one or two genotypes. LB resistance was assayed in the laboratory test (163 assessments). SCAR marker analysis covered 263 genotypes. Using this selection, 117 accessions representing 17 wild and cultivated *Solanum* species from seven *Petota* series were screened with SCAR markers of ten *Rpi* genes: *Rpi-R1*, *Rpi-R2*/*Rpi-blb3*, *Rpi-R3a*, *Rpi-R3b*, *Rpi-R8*, *Rpi-blb1*/*Rpi-sto1*, *Rpi-blb2,* and *Rpi-vnt1*. This communication also includes some data from our previous studies of the VIR collection and as of to date presents the most comprehensive description of this collection as regards LB resistance.

Because of broad race specificity, the *Rpi-blb1*/*Rpi-sto1*, *Rpi-blb2*, *Rpi-R2,* and *Rpi-vnt1* genes are currently most popular as the basis of durable LB resistance [49,50,51,52] and discovering new alleles of these genes would greatly benefit potato breeding. These genes became the focus of the present study. The patterns of plant response to *P. infestans* and the profiles of *Rpi* genes are discussed regarding the location of investigated species and their systematic position. Some data to be found below have been previously reported at several Euroblight workshops (https://agro.au.dk/forskning/internationale-platforme/euroblight/; assessed on 4 January 2023)

## 2. Results

### 2.1. LB Resistance

The frequency of resistant accessions (7–9 points by 9-point scale) varied from 0.11 to 0.71 depending on *Solanum* species (Table 1). The North American species *S. bulbocastanum*, *S. demissum,* and *S. cardiophyllum* Lindl. clearly predominated among the resistant forms. Among the South American species, the best was *S. berthaultii* Hawkes, with the frequency of resistant genotypes of 0.5.

LB resistance varied within species and even within accessions (between individual genotypes). In *S. bulbocastanum*, the accessions k-19981, k-21266, and k-21274 were resistant, k-24866 was susceptible, the accession k-24855 was weakly affected by artificial infection, whereas each of the accessions k-23174 and 25351 comprised both resistant and weakly affected forms (Appendix A). In *S. pinnatisectum* Dun., the accessions k-24239, k-24243, and 24415 were resistant; three accessions were moderately resistant or slightly affected by artificial infection (Appendix A). In *S. demissum*, most accessions were resistant in all experiments, though the accessions k-15174, k-15175, and k-24887 were slightly affected by the pathogen in all tests (Appendix A). In *S. berthaultii*, the accession k-19961 was resistant, in all other cases, plant response varied from weak damage to resistance. Similarly, in *S. verrucosum*, the accession k-24990 was resistant, the accessions k-24991 and k-24992 were susceptible, and in all other cases plants were moderately resistant to LB infection (Appendix A). Accessions of *S. stoloniferum* greatly vary in their LB response: we observed a considerable diversity of phenotypes within individual accessions k-3554, 24973 and high LB resistance in the accessions k-3360, k-21618 and k-24981 (Appendix A). The accessions of *S. venturii* k-12658, 25394, 25396, 25397, 25398 were obtained as seeds from the accessions CGN 17999, CGN 17998, CGN 18109, CGN 18279, and CGN 22703, respectively. Among them, the accessions κ-12658 and κ-25398 were resistant, whereas within the *S. venturii* accession k-25397, plants considerably differed in their response to infection: from susceptible to highly resistant. The variation within the accession κ-25394 was less prominent, from weakly affected to resistant (Appendix A).

### 2.2. Rpi Genes

The data for particular *Petota* accessions are combined in Appendix A; they are summed up per species and per series in Table 2 and Table 3.

As anticipated, the most prominent in the *Petota* set under study were the patterns of SCAR markers recognizing the *Rpi-R1*, *Rpi-2*, *Rpi-R3a*, *Rpi-3b*, and *Rpi-R8* genes, which were initially found in *S. demissum* [4,6,9,14]; below we will call these genes demissoid. In *S. demissum*, the marker frequencies for these genes varied from 0.18 (Rpi-R3a-1380) and 0.17 (Rpi-R8-1258) to 0.9 (Rpi-R3b-378). Many of these genes or their structural homologs were also recognized in other North American series at comparable frequencies (the species are listed in the descending frequency order): *Rpi-R1* in *S. stoloniferum* and *S. polytrichon*, *Rpi-R2* in *S. stoloniferum*, *S. polytrichon*, *S. bulbocastanum*, *S. pinnatisectum*, and *S. cardiophyllum*, *Rpi-R3a* in *S. cardiophyllum* and *S. cardiophyllum* ssp. ehrenbergii Bitt., *S. bulbocastanum* and *S. stoloniferum*, *Rpi-3b* in all *Pinnactisecta* species included in this study, as well as in *S. stoloniferum* and *S. bulbocastanum*, and *Rpi-R8* in *S. stoloniferum* (Table 2 and Table 3). Several *S. stoloniferum* accessions each comprised markers for two or even three demissoid genes; such plants were usually highly resistant to *P. infestans* (Appendix A). Notable is the presence of putative structural homologs of demissoid genes in the series *Tuberosa*: *Rpi-R2* in *S. avilesii* Hawkes et Hjerting, *S. venturii*, *S vernei* Bitt. et Wittm., *Rpi-R3a* in *S. microdontum* ssp. simplicifolium Bitt., *Rpi-3b* in *S. venturii* and *S. phureja* Juz. et Buk. and *Rpi-R8* in many *Tuberosa* species. Nevertheless, the functional activity of these genes, especially beyond *S. demissum*, will demand further investigation.

In *S. bulbocastanum*, we observe a high frequency of the markers for three *Rpi* genes initially discovered in this potato species: Rpi-blb1-821 (0.48), Rpi-blb2-976 (0.8) and Rpi-blb3-305 (0.75). The frequency of Rpi-sto1-890, the marker of *Rpi-sto1* orthologous to *Rpi-blb1*, was 0.53 in *S. bulbocastanum*, similar to the frequency of the marker Rpi-blb1-821. Most *S. bulbocastanum* plants containing these markers were highly resistant to *P. infestans* (Appendix A): however, our marker analysis did not include the susceptible accession k-23181 (two genotypes). In *S. bulbocastanum*, we also found a high frequency of the markers for demissoid genes *Rpi-R2*, *Rpi-3a*, and *Rpi-3b*. 

To screen for the *Rpi-blb1* gene, we usually employ several markers covering different regions of this gene. The marker of this gene, Rpi-blb1-821, and the marker of its orthologue, Rpi-sto1-890, which are located widely apart on the gene sequence, were tightly linked in *S. bulbocastanum* accessions and two *Longipedicellata* accessions (Appendix A). The sequences of *Rpi-blb1/Rpi-sto1* amplicons corresponding to these markers (the NCBI Genbank accessions KP317986–KP317990) were 100% identical to the prototype genes AY426259 and EU884421 [53]. In this study, we also discovered both markers, Rpi-blb1-821 and Rpi-sto1-890, in the *S. cardiophyllum* accession k-24203-435 (Appendix A). The fragments cloned from this accession were 99% identical to the prototype genes and differed by several SNPs (see also [53]). The function of these fragments has not yet been established.

In addition to *S. bulbocastanum*, the marker for *Rpi-blb2* was found in *S. cardiophyllum* (Table 2). The cloned amplicon differed from the prototype gene from *S. bulbocastanum* by several insertions. Less expected putative homologs of this gene in *S. alandiae* Card. (OP903197), *S. okadae* Hawkes et Hjerting and *S. microdontum* (Table 3; see also [54]) await further elucidation. In particular, the *Rpi-blb2* amplicon cloned from *S. alandiae* differed from its *S. bulbocastanum* prototype by deletion and several substitutions and was apparently a dysfunctional homolog [54].

The *Rpi-R2* gene from *S. demissum* and *S. schenckii* (FJ536325 and GU563975) is 96–99% identical to its ortholog *Rpi-blb3* from *S. bulbocastanum* (FJ536346) [39]. The markers of *Rpi-blb3* and *Rpi-R2* cover different regions of these orthologous sequences and do not always match in the three *Solanum* series. Most important, the frequencies of the two markers considerably diverge in different species (Table 2). We, therefore, presume that many individual plants, foremost in *Bulbocastana* and also in *Longipedicillata*, *Pinnatisecta*, and probably *Tuberosa* (e.g., *S. venturii*), each comprise both orthologs (Appendix A). Among the *Tuberosa* species, *Rpi-R2* homologs from *S. alandiae* (OP 749981 and OP 903199) and *S. okadae* were 94–97% identical to the prototype genes from *S. demissum* and *S. bulbocastanum*. Characteristically, the homologs from *S. bulbocastanum*, *S. alandiae*, and *S. okadae* differed from the *demissum* prototype *Rpi-R2 gene* by several SNPs and a six-nucleotide insertion [54]. In the case of *S cardiophyllum* and *S. cardiophyllum* ssp. *ehrenbergii*, we registered the different frequencies of the *Rpi-R2* gene marker (χ2 = 5.5, *p* < 0.02).

To our surprise, the frequencies of the marker for the *Rpi-R8* gene in many *Tuberosa* species considerably exceeded those in *S. demissum* and *S. stoloniferum* (Table 2). The fragments of the *Rpi-R8* gene cloned from *S. alandiae* (OP762711) and *S. okadae* [54] were 99% identical to the corresponding region in the prototype gene in *S. demissum* (KU530153). Two former sequences differed by several SNPs.

Diploid *S. bulbocastanum* and polyploid North American species significantly differed in the frequencies of *Rpi* gene markers. Such is the case of the *Rpi-R1* gene in *S. bulbocastanum* vs. *S. demissum* (χ2 = 20.5, *p* < 0.001) and vs. *S. stoloniferum* (χ2 = 10.4, *p* < 0.002). In the case of the *Rpi-3b*, *Rpi-R8*, and *Rpi-blb2* genes, the frequencies of the markers in *S. bulbocastanum* significantly differed from those in *S. stoloniferum* (χ2 = 7.0, *p* = 0.008; χ2 = 8.5, *p* = 0.004; χ2 = 8.9, *p* = 0.003, respectively).

The marker of *Rpi-vnt1* was found in all accessions of *S. bulbocastanum* and *S. stoloniferum* and absent from *S. demissum*, with the sole exception of the accession k-19997; in this aspect, marker frequencies in two former species significantly differed (χ2 = 16.4, *p* < 0.001, χ2 = 13.8, *p* < 0.001, respectively). The sequences from *S. bulbocastanum* (PI255516, PI275194), *S. cardiophyllum* (PI283062), and *S. stoloniferum* (PI195169, k-24420) were 92–96% identical to the prototype gene and differed from it by 5–9 SNPs.

## 3. General Discussion and Conclusions

### 3.1. Polymorphisms of Wild Potatoes in the VIR Collection

In the course of long evolution of tuber-bearing potatoes, the initial whole genome duplication and allopolyploidization by interspecific hybridization were followed by extensive gene diversification during *Solanum* speciation and plant adjustment to highly diverse habitats in North and South America. The characteristic case of series *Tuberosa* also included the adaptation to pathogens and selection under the pressure of informed breeding and cultivation. In particular, on the basis of his studies of LB-resistant potato forms, Budin [55] identified four centers of potato breeding: the Mexican, Colombian-Ecuadorian, Bolivian and Argentinean, which coincided with the natural habitats of *P. infestans-*resistant forms under the climates most favorable for pathogen dissemination. Sometimes new environments have been widely different from those in the areas of initial potato domestication and cultivation. All these processes of further gene evolution comprised tandem duplications, intergenic and intragenic recombination, selective retention, and neofunctionalization of retained key loci [56,57,58,59,60,61,62,63]. As an outcome, we observe a multihued landscape of agronomically important genes in the cultivated *Solanum* species and landraces along with the structural homologs of these genes of yet unknown functions in the wild species, especially in the aboriginal forms confined to the South American habitats and maintained in the local genetic collections [64]. The evolutionary analyses of this landscape would substantially promote and accelerate the search for the variation sought after by both evolution researchers and breeders [65]. This context helps elucidate the evolution of *Rpi* genes, assess their present diversity, and grasp the prospects to discover new beneficial alleles both in genetic collections of wild species [10,25,66] and in already existing multi-parent hybrids, which accumulated germplasm from numerous wild species introgressed sometimes many years ago [52].

The diversity of potatoes and their wild relatives has been repeatedly discussed against different taxonomies of the *Petota* section as the latter was revised to match the volumes of series and species and to include new, previously unknown species. The species systems were developed by Correll [67], Bukasov [46], and Hawkes [48]. Gorbatenko [47] took into account new species discovered at the end of the 20th century in Peru and Bolivia to develop the species system for South America. To classify potato species growing in the North, Central, and southern part of South America, Spooner et al. [68] combined the classical morphophysiological approach with the evidence obtained with several molecular technologies. Current systems of the *Petota* section widely differ in the number and composition of supraspecific taxa (series according to the classification by Correll, Bukasov, Hawkes, and Gorbatenko vs. groups according to the classification by Spooner) as well as in the number and grouping of species.

These revisions in the *Petota* taxonomy have split the ex situ potato world. While such depositary as the International Potato Center (CIP) organizes its collection according to the taxonomy by Hawkes, other major potato genebanks, such as the USDA-ARS Potato Genebank NRSP-6, adopted the classification by Spooner. Potato curators in all genebanks acknowledge the revision by Spooner as a significant advance in the field, yet the taxa by Hawkes are employed to manage their ex situ collections [69].

On the basis of early cytogenetic data, the genomes of most *Petota* species explored in this study are AA and AAAA in diploid and autotetrapoid *Tuberosa*, BB in diploid *Bulbocastana* and *Pinnatisecta* and AABB in allotetraploid *Longipedicellata*. Two hypotheses based on classical cytogenetic data and genomic in situ hybridization presume that the genome A in *Longipedicellata* originated either from *S. chacoense* or *S. verrucosum* (see [48,68] for detailed discussion); however, both hypotheses have not been extensively tested with the latest molecular methods. The systematic affiliation of three genomes of *S. demissum* has also been under active discussion [68]. Hopefully, these genome disputes will be finally resolved as more and more *Petota* genomes are completely deciphered [70,71,72,73,74,75,76].

The study of the VIR potato collection was started in 1921 by S.M. Bukasov and his colleagues and considerably expanded when wild potato tubers collected by the VIR expeditions to two American continents started to arrive in 1927 [77]. The present VIR collection maintains 8150 *Petota* accessions, including 1990 accessions of wild species, 3200 accessions of cultivated species, 2360 cultivars, and 600 advanced clones [45]. From the very beginning, the work with the collection has focused on mobilizing the germplasm pool of potatoes and related species and the genetic studies of these resources, as well as selecting the most valuable plant accessions and developing new sources for potato breeding. For this study, we selected Petota series and species from two dissimilar potato groups representing the North American and South American centers of genetic diversity. We focused on the accessions that produced tubers under our experimental conditions; thus, our study could proceed for several years.

### 3.2. LB Resistance

The VIR accessions of *Petota* species have been repeatedly assessed for LB resistance, and new sources of LB resistance were identified [20,53,77,78,79]. Leaf LB resistance is characteristic of the North American species from series *Demissa* (*S. demissum*, *S. guerreroense* Corr. and *S. hougasii* Corr.), *Pinnatisecta* (*S. pinnatisectum* and *S. tarnii* Hawkes et Hjerting), *S. bulbocastanum* and *S. stoloniferum* and the South American species *S. vernei* and *S. berthaultii*

In several cases, the assessments of LB resistance of the potato accessions in the VIR collection can be matched up against the evidence reported by other authors. We compared our data to the evidence from the laboratory tests carried out in the US Genbank (these evaluation data are provided by the Germplasm Resource Information Network (GRIN) database (https://npgsweb.ars-grin.gov/gringlobal/search; assessed on 4 January 2023).). An excellent agreement was established for 94 accessions of wild and cultivated potato species from two collections: Spearman’s correlation coefficient was 0.76 (*p* = 0.05). Another illustration is the LB resistance of *S. venturii*: our evidence is in good agreement with the data for the accessions CGN 18279, CGN 22703 and CGN 18109 reported by Foster et al. [40]. When Bachman-Pfabe et al. [29] tested the potato collection from the IPK Genebank (https://www.ipk-gatersleben.de/en/research/genebank; assessed on 4 January 2023).) for tuber resistance to *P. infestans*, the most prominent, similar to the present study, were the accessions of *S. bulbocastanum*, *S. pinnatisectum*, and *S. stoloniferum*.

### 3.3. Diversity of the Rpi Genes

A considerable part of potato genomes is represented by the genes that determine resistance to numerous diseases and pests. Among them, the dominant *Rpi* genes apparently evolved from the NB-LRR predecessor structures (protogenes) when the *Solanum* species were adjusting to the particular habitats in both American continents and co-evolving with the local repertoires of *P. infestans* [9,80,81,82,83].

Recent advances in the phylogenetic and phylogeographic studies of this diversity of *Rpi* genes in the numerous wild and cultivated *Solanum* species and landraces also owe much to the sequencing of complete genomes [70,71,72,73,74,75,76] and transcriptomes [72,84,85,86]. The big data from sequencing are supplemented with the evidence obtained by extensive screening of nuclear DNA with various anonymous and gene-specific markers [3,34,57,68,75,87,88,89,90,91]. Another kind of evidence is obtained by researching into the polymorphisms of individual genes and rapidly evolving clusters of NB-LRR resistance genes [14,59,80,83,92].

In hexaploid *S. demissum*, only one haplotype (genome) comprised the functional *Rpi-R1* gene, and its structural homologs (pseudogenes) in two other genomes might act as a reservoir of gene sequences for further evolution [92]. The distribution of other *Rpi* genes between haplotypes of *S. demissum* has not been explicitly established.

The gene *Rpi-blb1* was firmly established as the marker of genome B in the series *Bulbocastana* and *Longipedicellata* rather than *Pinnatisecta* [3,53]. There are, however, several exceptions. Lokossou. [39] found the marker Rpi-blb1-821 in two Mexican accessions of *S. cardiophyllum* (CGN 18326 and CGN 22387). Similar evidence is reported in this communication (Appendix A). Tiwari et al. [33] registered an 846-bp homolog (KJ472309) from *S. cardiophyllum* with 82% identity to the prototype *Rpi-blb1* gene. Previously Pankin et al. [93,94,95] cloned from *S. cardiophyllum* several DNA fragments corresponding to the CC region of the *Rpi-blb1* gene; among them, two sequences over 1 kb in length (JN688099 and JN698894), cloned from the accession k16828-390 were 94% similar to the prototype gene. In addition to *S. cardiophyllum*, a full-length homolog of the *Rpi-blb1* gene was found in the genome of *S. pinnatisectum* [75], its sequence (CP047564) is 84% identical to *Rpi-blb1*. Fadina et al. [53] did not find the SCAR markers of the *Rpi-blb1/Rpi-sto1* genes in the accessions of *Demissa* and *Pinnatisecta* (with exception of a single *S. cardiophyllum* accession k-24207, GLKS2152, 420). The markers of *Rpi-blb1/Rpi-sto1* were also absent from resistant *S. verrucosum* accessions comprising the *RBver* homolog [96].

Other *Rpi* genes in the wild *Petota* collection maintained in the VIR Genbank have been previously researched in our laboratory. Sokolova et al. [32] screened over 200 wild *Solanum* accessions (half of them from the VIR collection) representing 21 species of six *Petota* series with the SCAR markers for the *Rpi-R1* and *Rpi-R3a* genes. As a whole, these genes were restricted to the Mexican species: the *Rpi-R1* gene was reported in the series *Demissa* and *Longipedicellata*, whereas the *Rpi-R3a* gene was also found in several accessions of *S. bulbocastanum*, *S. cardiophyllum*, and *S. cardiophyllum* ssp. ehrenbergii. *S. microdontum* was the only South American species comprising *Rpi-R3a*. Later the same researchers reported the SCAR marker of the *Rpi-R3b* gene in many *Pinnatisecta* accessions [97]. Beketova et al. [98] screened numerous accessions of *S. demissum*, *S. stoloniferum*, *S. papita*, and *S. polytrichon* (two latter species are included in *S. stoloniferum* sensu Spooner) from the VIR collection using three *Rpi-R1* markers. With the marker Rpi-R1-1205, they found the gene and/or its putative homologs in all these species. Next, more specific markers Rpi-R1-517 and Rpi-R1-513 were employed to discriminate between the *Rpi-R1* alleles putatively characteristic of *S. demissum* and *S. stoloniferum*, respectively. The deduced amino acid sequences of two alleles differed by three substitutions. The marker Rpi-R1-517 was found in all *S. demissum* accessions, whereas the presumable *stoloniferum* allele recognized with the marker Rpi-R1-513 was registered only in some *stoloniferum* accessions. Zoteeva et al. [78] reported the marker of the *Rpi-R3a* gene in the Mexican species *S. guerreroense* (Demissa, Iopetala group), accession k-18407.

Foster et al. [40] and Pel et al. [41] identified three functional alleles of the *Rpi-vnt1* gene in *S. venturiii*. Using allele mining with specific PCR primers, Pel [99] recognized the functional and non-functional homologs of this gene in many South American species. The list of species comprising these homologs was recently complemented with *S. alandiae* (OP889280) and *S. okadae* (OP903198) [62] and with *S. phureia* and *S. stenotomum* [100]. In the current study, the marker of the *Rpi-vnt1* gene was also registered in two *Tuberosa* species absent from the Pel’s list: *S. alandiae* and *S. berthaultii* (Table 2). Pel [99] presumed that the *Rpi-vnt1* gene is restricted to the series *Tuberosa*; however, Tiwari et al. [75] reported a structural homolog of *Rpi-vnt1* (CP047560) on chromosome 9 in the recently sequenced genome of *S. pinnatisectum* Dun. (CGN17745). In the present study, the marker Rpi-vnt1-612 was found in other North American species: *S. bulbocastanum*, *S. cardiophyllum*, and *S. stoloniferum* and even in one accession of *S. demissum* (Table 2; Appendix A). The frequencies of this marker in the Mesoamerican species (Table 2) exceeded those reported for the South American species [99]. Remarkably, the only North American species from the series *Tuberosa*, *S. verrucosum*, was devoid of this marker (Table 2).

When we compared the *Rpi-vnt1* homologs obtained in this study with the prototype gene and its numerous homologs (pseudogenes) (Appendix A), they were distinctly divided into two clusters: the translated and most probably functional genes and the pseudogenes (Appendix A). Cluster 1 of translated sequences is subdivided into two subclusters. The subcluster 1a comprises the prototype alleles *Rpi-vnt1.1*, *Rpi-vnt1.2*, and *Rpi-vnt1.3* (vnt_FJ423044, vnt_FJ423045, vnt_FJ423046) and *Rpi-vnt1* sequence tub_OP617268 from *S. tuberosum* cultivar Alouette (this cultivar contains the *Rpi-vnt1-3* allele [36]) and also non-translatable sequences of *Rpi-vnt1.1* pseudogenes from *S. venturii* (vnt__GU386358), from S. tarijense (tar_GU338324 and tar_GU338326) and from *S. microdontum* ssp. *gigantophyllum* (mcd_GU338312). These pseudogenes are 94–97% identical to the prototype gene. The pseudogene from *S. venturii* (GU386358) comprises a 3-bp insertion encoding a stop codon. Subcluster 1b comprises the translatable sequences from the North American species *S. bulbocastanum*, *S. cardiophyllum*, and *S. stoloniferum* cloned in this study, with the identity to the prototype gene of 92–96%. These sequences contain a characteristic deletion in position 43–47 bp, which restores the proper reading frame. The region of 1–42 bp in *Rpi-vnt1* from the North American species is quite different from all other homologs of this gene in the NCBI Genebank. The second cluster in Appendix A combines pseudogenes found by Pel [99] in the South American Tuberosa species. Three outgroup sequences are 86–93% identical to the prototype gene.

The divergent evolution and specialization of the *Rpi-blb1* protogene(s) in *Petota* species left the functional homologs considerably different from *Rpi-blb1/Rpi-sto1*, such as *Rpi-bt1* in *S. bulbocastanum* (with the identity of 87% to *Rpi-blb1*) and *RBver* in *S. verrucosum* (the identity of 89%); several polymorphisms prevent recognition of these genes by the marker Rpi-blb1-821 [53]. NCBI Genebank database comprises many *Tuberosa* homologs of this gene with an identity below 90%. Notably, when using SCAR markers other than Rpi-blb1-821 and Rpi-sto1-890, many footprints of the *Rpi-blb1* gene of unknown functions were reported in diverse *Solanum* and even *Capsicum* species. While in *S. bulbocastanum* these genome fragments were linked to resistance to *P. infestans*, in other species such a relationship was not observed [33,93,94]. The structural homologs of *Rpi-vnt1* initially identified in *S. venturii* were found in many South and North American species ([54,99,100] and this communication); most probably they are pseudogenes.

Quite apart stands *S. verrucosum*; this North American species is combined by taxonomists in one clade with the southern South American *Tuberosa* [48,101,102]. Such treatment is not recognized universally, and Bukasov [46] even put this species into the separate series *Verrucosa*. Several accessions of *S. verrucosum* are highly resistant to *P. infestans*, and here Liu and Halterman [96] discovered the functional homologs of *Rpi-blb1*; these *RBver* sequences shared up to 89% nucleotide identity with the former gene. Using the specific *Avr* genes, these investigators demonstrated that most *S. verrucosum* accessions resistant to *P. infestans* also comprised the functional homologs of the *Rpi-R8* and/or *Rpi-R9* genes [103]. More recently, Chen et al. [44] applied two complementary enrichment strategies that targeted resistance genes (RenSeq) and single/low-copy number genes (generic-mapping enrichment sequencing; GenSeq), respectively, to investigate *Rpi* genes of *S. verrucosum*. Here, on the distal end of chromosome 9 rich in the NB-LRR sequences [80], many *Rpi* genes have previously been identified, including *Rpi-vnt1* from *S. venturii* [40,41]. The new gene, named *Rpi-ver1*, was different from the well-known set of *demissum* genes *Rpi-R1–Rpi-R11*, the *bulbocastanum* genes *Rpi-blb1* and *Rpi-blb2* and also from the *Rpi-vnt1* gene.

In *S. demissum* (k-15175, k-19997) and *S. stoloniferum* (k-3360, 21616, 23652, 24420, 24981, 24972, 24976), we registered SCAR markers for several *Rpi* genes per accession. Two accessions of *S. stoloniferum* k-24263 and k-24981 combined the markers for five genes (Appendix A). Two accessions of *S. stoloniferum* k-3554 and k-23652 combined the markers for three demissoid genes (Appendix A). The cases of several *Rpi* genes in one plant are especially promising sources for pyramiding/stacking technologies of breeding for broad-spectrum and durable disease resistances, including potato LB resistance [42,49,52].

In marker analysis, we find numerous homologs of specific *Rpi* genes in various species of the section *Petota*. The question arises as to their origin. One reason for the presence of “alien” homologs with an identity of over 90% is a cross-pollination of wild *Solanum* species with their cultivated and wild relatives. The expanded wild introgressions following polyploidy brought in alleles from outside of the geographic origin of particular species. As reported by Hardigan et al. [58] and Bethke et al. [23], 73% of alleles characteristic of wild species are found in North American potato varieties. Accessions of the same species may widely vary when they were collected from different habitats [10]. Another reason may be the divergent evolution of protogens: the structures that preceded the differentiation of the genomes of tuberous potato species went through a series of mutations. Functional *Rpi* genes arose during the further evolution of *Solanum* species, when, in the course of dispersal on the American mainland, the species adapted to different habitats with local pathogen repertoires. Recent studies show that the recombination level at resistance gene clusters is increased following pathogen infection, suggesting a mechanism that induces temporary genome instability in response to extremal stress conditions [39,82].

One way to characterize putative protogens is to study *Rpi* homologs in susceptible *Tuberosa* species with well-characterized genomes. The Solynthus potato variety [104] was bred to a unique level of homozygosity. Its genome has been fully sequenced, but the analysis of its individual genes is just beginning. Nevertheless, the BLAST analysis revealed in this genome the structural homologs of all the genes studied in our work; these homologs are 87–95% similar to the prototype genes. As a caveat, one should note that even after nine cycles of selfing, Solynthus has noticeable heterozygosity, possibly due to the preferential selection of heterozygous plants in the selection process [104].

Notwithstanding the ongoing progress in gene mining and allele identification, the attempts to directly link the *Rpi* gene activities to plant LB resistance have not been always convincing, even when gene functions were established by effectoromics and genetic transformation of plants susceptible to *P. infestans*. The situation is especially complicated in plants combining several *Rpi* genes of different specificity toward *P. infestans* races. Thus, in our case, the relationship between the presence of *Rpi* genes and plant resistance to *P. infestans* is qualitatively visible although not statistically significant, especially in *S. bulbocastanum*. In some cases, such a lack of significant relationship presumes that highly resistant plants comprise *Rpi* genes other than scored presently with our markers. Apparently, these *Solanum* species have not been adequately studied by molecular methods: hopefully, such potato accessions contain as yet unknown *Rpi* genes or new alleles of known *Rpi* genes. We believe that this problem will be overcome by using the new screening technologies directly recognizing the functional alleles of the *Rpi* genes.

The characteristic dichotomy in the geographic distribution of the *Rpi* genes may help elucidate their evolution in tuber-bearing potatoes. Our evidence confirmed the diverse location of the functional *Rpi* gene in two American continents. The structural homologs of the *Rpi-R2*, *Rpi-R3a*, *Rpi-R3b*, and *Rpi-R8* genes were found in the North American species other than *S. demissum*, the very species which was the original source of these genes for early potato breeding, and in some cases, in the South American *Tuberosa* species. In contrast, the *Rpi-blb1/Rpi-sto1* orthologs from *S. bulbocastanum* and *S. stoloniferum* were restricted to the Mesoamerican series *Bulbocastana*, *Longipedicellata* and possibly *Pinnatisecta*. Particularly interesting are the structural homologs of *Rpi-vnt1*. The functional gene was initially identified in *S. venturii*, and its homologs were found in many South American species, whereas here we report these homologs, for the first time, in the North American *Petota* series. Future studies will probably reveal whether such homologs are footprints of NB-LRR gene functionalization in the expanding area of tuber-bearing potatoes rather than protogenes rejected by *Rpi* gene specialization in particular disease landscapes.

## 4. Materials and Methods

Clonal collections of wild *Solanum* species were developed from seed accessions obtained from the N.I. Vavilov Institute of Plant Genetic Resources (VIR), Russia (http://www.vir.nw.ru/ assessed on 4 January 2023), the US Potato Genebank NRSP-6 (https://npgsweb.ars-grin.gov/gringlobal/search; assessed on 4 January 2023), CIP (International Potato Center, https://cipotato.org/ assessed on 4 January 2023) and the CGN potato collection (https://www.wur.nl/en/research-results/statutory-research-tasks/centre-for-genetic-resources-the-netherlands-1/plant-genetic-resources/genebank/cgn-crop-collections/cgn-potato-collection.htm; assessed on 4 January 2023) and were further maintained in VIR as tuber progenies (Appendix A).

Resistance to *P. infestans* was assayed in the laboratory test with detached leaves according to the Eucablight protocol (www.euroblight.net/; assessed on 4 January 2023) see also [105]). In this test, we employed a highly virulent and aggressive isolate of *P. infestans* N161 collected in the Moscow region and maintained in the Institute of Phytopathology (mating type A1; race 1,2,3,4,5,6,7,8,9,10,11; virulence genes *avr1*, *Avr2K*, *Avr2-likeMI*, *avr3aEM*, *avr4*, *Avr8*, *Avr-Smira1* (I, II), *Avr-blb1* (I, II), *Avr-blb2*, *Avr-vnt1* [106]) and potato variety Santé as a standard. The resistance indices of individual accessions and clones which were also screened with the SCAR markers of ten *Rpi* genes are listed in Appendix A.

Specific SCAR markers of *Rpi* genes employed in screening wild *Solanum* accessions were designed by other authors and verified against the prototype *Rpi* genes characteristic of particular *Solanum* species (Table 4).

When discerned in other *Petota* series, DNA fragments amplified with these markers are referred to as structural homologs rather than orthologues of the prototype genes. In several cases, to evaluate the structural relationship with the prototype genes, the amplicons from remote species were cloned using pGEMT Easy Vector System I (Promega, Madison, WI, USA) or pAL2-T vector (Evrogen, Moscow, Russia; https://evrogen.ru/; assessed on 4 January 2023) and sequenced with a nucleic acid analyzer ABI PRISM 3130xl (Applied Biosystems, Foster City, CA, USA) in the Institute of Agricultural Biotechnology or with an ABI PRISM 3500 (Applied Biosystems) in the Evrogen. BLAST 2.0.13.0 (https://blast.ncbi.nlm.nih.gov/Blast.cgi?PROGRAM=blastn&PAGE_TYPE=BlastSearch&LINK_LOC=blasthome; assessed on 4 January 2023) was used to mine genomic databases for *Rpi* genes and their homologs. SeqMan, Lasergene 7.0 programs, and MEGA version 10.2.1 [112] were employed to assemble sequence fragments.

The data were statistically processed by the parametric and nonparametric statistics methods using the Statistica StatSoft 13 software package (http://statsoft.ru/resources/support/new-features-statistica-13.php; assessed on 4 January 2023).

## Figures and Tables

**Table 1 plants-12-00273-t001:** Resistance of wild potato species accessions to *P. infestans* in the laboratory test.

Series	Species	Frequency of Resistant Accessions * (the Total Number)	The Average Score
*Bulbocastana*	*S. bulbocastanum*	0.71 (14)	6.7
*Pinnatisecta*	*S. pinnatisectum*	0.50 (6)	6.0
	*S. cardiophyllum*	0.60 (5)	6.4
	*S. cardiophyllum* ssp. *ehrenbergii*	0.17 (12)	4.7
	*S. jamesii*	0.11 (9)	5.1
	*S. stenophyllidium*	0 (2)	3.0
*Yungasensa*	*S. chacoense*	0.33 (6)	4.8
*Maglia*	*S. maglia*	n.d.**	
*Tuberosa (wild)*	*S. alandiae*	0 (2)	4.5
	*S. avilesii*	0.5 (2)	5.5
	*S. berthaultii*	0.50 (8)	6.5
	*S. microdontum*	0.20 (5)	5.8
	*S. microdontum* ssp. *gigantophyllum*, syn. *simplicifolium*	0.20 (5)	5.8
	*S. sucrense*	n.d.	
	*S. venturii*	0.29 (17)	6.6
	*S. vernei*	0 (5)	5.0
	*S. verrucosum*	0.37 (8)	5.4
*Tuberosa (cultivated)*	*S. phureja*	0 (3)	4.3
	*S. stenotomum*	0 (2)	6.4
*Demissa*	*S. demissum*	0.58 (12)	6.7
*Longipedicellata*	*S. stoloniferum*	0.25 (20)	5.4
	*S. polytrichon*	0 (13)	4.4

* 7–9 points by the 1–9 resistance scale (1, susceptible; 9, resistant); ** n.d.—no data.

**Table 2 plants-12-00273-t002:** Screening *Solanum* species with SCAR markers of the *Rpi* genes. Frequency of the markers (the number of accessions analyzed).

Series	Species	*Rpi-R1*	*Rpi-R2/Rpi-blb3*		*Rpi-R3a*	*Rpi-R3b*	*Rpi-R8*	*Rpi-blb1/Rpi-sto1*		*Rpi-blb2*	*Rpi-vnt1*
		Markers (the Numbers of Scored Accession Are Shown in Parenthesis)									
		Rpi-R1-1205	Rpi-R2-1137	Rpi-blb3-305	Rpi-R3a-1380	Rpi-R3b-378	Rpi-R8-1258	Rpi-blb1-821	Rpi-sto1-890	Rpi-blb2- 976	Rpi-vnt1-612
*Bulbocastana*	*S. bulbocastanum*	0 (34)	0.37 (16)	0.75 (16)	0.29 (34)	0.42(24)	0 (13)	0.48 (25)	0.53 (15)	0.80 (15)	1.0 (15)
*Pinnatisecta*	*S. pinnatisectum*	0 (9)	0.5 (6)	0.28 (7)	0 (9)	0.33 (6)	0 (6)	0 (6)	0 (6)	0 (6)	1.0 (6)
	*S. cardiophyllum*	0 (7)	0.14 (7)	0.28 (7)	0.28 (7)	0.17 or 1.0 (6)	0 (6)	0.17 (6)	0.17 (6)	0.33 (6)	0.66 (6)
	*S. cardiophyllum* ssp. *ehrenbergii*	0 (15)	0.75 (8)	0.12 (8)	0.26 (15)	0.78 (14)	0 (4)	0 (12)	0 (8)	0.14 (7)	1 (4)
	*S. jamesii*	0 (12)	0 (1)	0 (1)	0 (12)	0.37 (8)	n.d.	0 (2)	0 (1)	0 (1)	n.d.
	*S. stenophyllidium*	0 (3)	1.0 (2)	0 (2)	0 (3)	0.33 (3)	n.d.	0 (2)	0 (2)	0 (2)	n.d.
*Yungasensa*	*S. chacoense*	0 (11)	0 (11)	0 (11)	0 (11)	0.36 (11)	0 (11)	0 (11)	0 (11)	0 (11)	0 (11)
*Maglia*	*S. maglia*	0 (3)	0 (3)	0 (3)	0 (3)	0 (3)	1 (3)	0 (3)	0 (3)	0 (3)	0 (3)
*Tuberosa (wild)*	*S. alandiae*	0 (5)	0 (4)	0 (4)	0 (5)	0 (3)	0.6 (5)	0 (4)	0 (4)	0.8 (5)	0.2 (5)
	*S. avilesii*	0 (2)	0.5 (2)	0 (2)	0 (2)	0 (2)	1 (2)	0 (2)	0 (2)	1.0 (2)	1.0 (2)
	*S. berthaultii*	0.09 (11)	0 (5)	0 (5)	0 (11)	0 (6)	1 (4)	0 (5)	0 (5)	0.4 (5)	0.6 (5)
	*S. microdontum*	0 (9)	0 (8)	0 (8)	0 (8)	0 (8)	0.43 (7)	0 (8)	0 (7)	0 (7)	0.37 (7)
	*S. microdontum* ssp. *gigantophyllum*, syn. *simplicifolium*	0 (4)	0 (3)	0 (3)	0.25(4)	0 (3)	0.33 (3)	0 (3)	0 (2)	0.33 (1)	0.66 (3)
	*S. sucrense*	0 (2)	0 (2)	0 (2)	0 (2)	0 (2)	0.5 (2)	0 (2)	0 (1)	0 (2)	0.5 (2)
	*S. venturii*	0 (6)	0.17 (6)	0.17 (6)	0 (6)	0.17 (6)	0.83 (6)	0 (6)	0 (6)	0.33 (6)	1.0 (6)
	*S. vernei*	0 (5)	0.2 (5)	0 (5)	0 (5)	0 (5)	0.6 (5)	0 (5)	0 (5)	0 (4)	0.4 (5)
	*S. verrucosum*	0 (15)	0.08 (12)	0 (12)	0 (15)	0.07 (14)	0 (12)	0 (14)	0 (12)	0 (12)	0 (12)
*Tuberosa (cultivated)*	*S. phureja*	0 (6)	0 (6)	0 (6)	0 (6)	0.33 (6)	0.66 (6)	0 (6)	0 (6)	0 (6)	0.66 (6)
	*S. stenotomum*	0 (2)	0 (2)	0 (2)	0 (2)	0 (2)	0.5 (2)	0 (2)	0 (2)	0 (2)	0.5 (2)
*Demissa*	*S. demissum*	0.46 (37)	0.5 (6)	0.33 (6)	0.18 (38)	0.9 (10)	0.17(6)	0 (8)	0 (6)	0 (6)	0.17 (6)
*Longipedicellata*	*S. stoloniferum*	0.26 (46)	0.67 (18)	0.5 (18)	0.2 (46)	0.79 (24)	0.5 (12)	0.33 (24)	0.22 (18)	0.28 (18)	1.0 (12)
	*S. polytrichon*	0.17 (12)	1.0 (5)	0.2 (5)	0.08 (12)	0.33 (9)	0.33 (3)	0.28 (7)	0 (5)	0.4 (5)	0.67 (3)

**Table 3 plants-12-00273-t003:** Screening *Solanum* species from the seven taxonomic series of Section *Petota* Dum. with SCAR markers of the *Rpi* genes. Frequency of the markers (the number of accessions analyzed).

Series	*Rpi-R1*	*Rpi-R2/Rpi-blb3*		*Rpi-R3a*	*Rpi-R3b*	*Rpi-R8*	*Rpi-blb1/Rpi-sto1*		*Rpi-blb2*	*Rpi-vnt1*
	Markers (the Numbers of Scored Accession Are Shown in Parenthesis)									
	Rpi-R1-1205	Rpi-R2-1137	Rpi-Rpi-blb3-305	Rpi-R3a-1380	Rpi-R3b-378	Rpi-R8-1258	Rpi-blb-1-821	Rpi-sto1-890	Rpi-blb2-976	Rpi-vnt1-612
*Bulbocastana*	0 (34)	0.37 (16)	0.75 (16)	0.29 (34)	0.42(24)	0 (13)	0.48 (25)	0.53 (15)	0.80 (15)	1.0 (15)
*Pinnatisecta*	0 (46)	0.50 (24)	0.20 (25)	0.13 (46)	0.55 (31) *	0.06 (16)	0.03 (28)	0.04 (23)	0.14 (22)	0.87 (16)
*Yungasensa*	0 (11)	0 (11)	0 (11)	0 (11)	0.36 (11)	0 (11)	0 (11)	0 (11)	0 (11)	0 (11)
*Maglia*	0 (3)	0 (3)	0 (3)	0 (3)	0 (3)	1.0 (3)	0 (3)	0 (3)	0 (3)	0 (3)
*Tuberosa (wild)*	0.02 (62)	0.08 (50)	0.02 (50)	0.02 (61)	0.04 (50)	0.51 (49)	0 (52)	0 (48)	0.22 (49)	0.40 (50)
*Tuberosa (cultivated)*	0 (8)	0 (8)	0 (8)	0 (8)	0.25 (8)	0.62 (8)	0 (8)	0 (8)	0 (8)	0.62 (8)
*Demissa*	0.46 (37)	0.50 (6)	0.33 (6)	0.18 (38)	0.9 (10)	0.17 (6)	0 (8)	0 (6)	0 (6)	0.17 (6)
*Longipedicellata*	0.24 (58)	0.74 (23)	0.43 (23)	0.17 (58)	0.66 (33)	0.47 (15)	0.32 (31)	0.17 (23)	0.30 (23)	0.93 (15)

* without of *S. cardiophyllum.*

**Table 4 plants-12-00273-t004:** SCAR markers of the Rpi genes used in this study.

Gene	Solanum Species, Gene Accession Numbers in the NCBI GenBank	Marker, Length, bp	Marker Position on the Gene, bp	Anneal. Temp., °C	Primer Sequences	References
*RB/Rpi-blb1*	*S. bulbocastanum*, AY426259	Rpi-blb1-821	4989–5809	62	F-aacctgtatggcagtggcatg R-gtcagaaaagggcactcgtg	[3]
*Rpi-sto1*	*S. stoloniferum*, EU884421	Rpi-sto1-890	241–1130	65	F-accaaggccacaagattctc R-cctgcggttcggttaataca	[49,107]
*Rpi-blb2*	*S. bulbocastanum*, DQ122125	Rpi-blb2-976	4004–4979	58	F-ggactgggtaacgacaatcc R-atttatggctgcagaggacc	[108]
*Rpi-R1*	*S. demissum*, AF447489	Rpi-R1-1206	5126–6331	61	F-cactcgtgacatatcctcacta R-gtagtacctatcttatttctgcaagaat	[32]
*Rpi-R2*	*S. demissum*, FJ536325	Rpi-R2-1137	1277–2413	60	F-aagatcaagtggtaaaggctgatg R-atctttctagctaaagatcacg	[109]
*Rpi-blb3*	*S. bulbocastanum*, FJ536346	Rpi-blb3-305	5551–5855	63.5	F-agctttttgagtgtgtaattgg R-gtaactacggactcgaggg	[110]
*Rpi-R3a*	*S. demissum*, AY849382	Rpi-R3a-1380	1677–3056	64	F-tccgacatgtattgatctccctg R-agccacttcagcttcttacagtagg	[32]
*Rpi-R3b*	*S. demissum*, JF900492	Rpi-R3b-378	94,818–95,195	64	F-gtcgatgaatgctatgtttctcgaga R-accagtttcttgcaattccagattg	[111]
*Rpi-R8*	*S. demissum*, KU530153	Rpi-R8-1258	73,693–74,950	62.5	F-aacaagagatgaattaagtcggtagc R-gctgtaggtgcaatgttgaagga	[43] modif.
*Rpi-vnt1*	*S. venturii*, FJ423044–FJ423046	Rpi-vnt1-612	89–700	58	F-ccttcctcatcctcacatttag R-gcatgccaactattgaaacaac	[41]

## Data Availability

Not applicable.

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
