# Peer review of "Diversity of Late Blight Resistance Genes in the VIR Potato Collection"

_plants, 2023, doi:10.3390/plants12020273_

Round 1
Reviewer 1 Report (Previous Reviewer 1)
The authors made a partial improvement. The methodological and results part still remain problematic. The authors present a lot of data in the form of supplements, where at first glance there is a lot of data, but n.d. or "0" information appears (not detected, which DNA analysis can do in case of a new resistance variant), but it is not clear why this also appears in the resistance assessment. The difference in the evaluated genotypes is quite large (more than 20%). Therefore, it is problematic to generalize the results. In the case of uneven data distribution, the results are distorted. It is ideal to compare what was tested for resistance, but also DNA markers! The peculiarity is that the discussion is connected with the conclusion. These parts should be clearly separated. For the above reasons, I cannot recommend it for publication.
Author Response
Dear Reviewer 1,
Once again, let us thank most heartily you for the labors and constructive suggestions concerning the revised version of our paper.
Below we provide point-to-point responses to each of your comments.
The authors made a partial improvement. The methodological and results part still remain problematic. The authors present a lot of data in the form of supplements, where at first glance there is a lot of data, but n.d. or "0" information appears (not detected, which DNA analysis can do in case of a new resistance variant), but it is not clear why this also appears in the resistance assessment.
As far as we understand this comment, the shortage of data is a serious deficiency aggravating other defects of our MS. We completely agree to this criticism. However, when working with wild potato plants in three distant laboratories, the scarcity of initial material is also aggravated by the problems of logistics.
The difference in the evaluated genotypes is quite large (more than 20%). Therefore, it is problematic to generalize the results. In the case of uneven data distribution, the results are distorted. It is ideal to compare what was tested for resistance, but also DNA markers!
Wild Solanum accessions in the Genbanks demonstrate considerable intraspecies diversity regarding both the phenotypic resistance and the presence of SCAR markers. Wherever possible, we try overcoming this problem by developing clonal subcollections within the particular accessions.
The peculiarity is that the discussion is connected with the conclusion. These parts should be clearly separated.
The section General Discussion and Conclusion was constructed during MS revision following the advice by Editor and Reviewer 3.
Gratefully yours,
The Authors: Elena V. Rogozina, Alyona A. Gurina, Nadezhda A. Chalaya, Nadezhda M. Zoteyeva, Mariya A. Kuznetsova, Mariya P. Beketova, Oksana A. Muratova, Ekaterina A. Sokolova, Polina E. Drobyazina, and Emil E. Khavkin
December 23, 2022
Reviewer 2 Report (Previous Reviewer 2)
The authors attended the suggestions and it can be published.
Author Response
We thank the Reviewer 2 for the positive evaluation.
Gratefully yours,
The Authors: Elena V. Rogozina, Alyona A. Gurina, Nadezhda A. Chalaya, Nadezhda M. Zoteyeva, Mariya A. Kuznetsova, Mariya P. Beketova, Oksana A. Muratova, Ekaterina A. Sokolova, Polina E. Drobyazina, and Emil E. Khavkin
December 23, 2022
Reviewer 3 Report (Previous Reviewer 3)
I am happy that you have dealt with the issues I raised in my review of original manuscript. When proof reading:
L53 I think safety should be security
L307 North American (spelling mistake)
L494 I think that areal should be area
L510 Are small a's correct for avr1, avr3a and avr4?
References: Has something gone wrong with 1st reference?
Author Response
Dear Reviewer 3,
Once again, let us thank most heartily you for the labors and constructive suggestions concerning the revised version of our paper.
Below we provide point-to-point responses to each of your comments.
When proof reading:
L53 I think safety should be security
L307 North American (spelling mistake)
L494 I think that areal should be area
We made all necessary changes concerning these points.
References: Has something gone wrong with 1st reference?
It is correct
L510 Are small a's correct for avr1, avr3a and avr4?
avr/Avr denotes alleles that differ in virulence. For details, please see [107] and the references therein.
Gratefully yours,
The Authors: Elena V. Rogozina, Alyona A. Gurina, Nadezhda A. Chalaya, Nadezhda M. Zoteyeva, Mariya A. Kuznetsova, Mariya P. Beketova, Oksana A. Muratova, Ekaterina A. Sokolova, Polina E. Drobyazina, and Emil E. Khavkin
December 23, 2022
Round 2
Reviewer 1 Report (Previous Reviewer 1)
The authors themselves admit the weaknesses of the manuscript and the impossibility of dealing 100% with comments.
Min. I recommend checking the journal format carefully: line 666 vs. 770-771 vs. 850; 705-706; 762? 818; 841, etc.
Author Response
Dear Reviewer 1,
We thank you for the labors and constructive suggestions concerning the revised version of our paper.
As you recommend to check the journal format carefully and look line 666 vs. 770-771 vs. 850; 705-706; 762? 818; 841, etc.
we checked References and made corrections.
Gratefully yours,
The Authors: Elena V. Rogozina, Alyona A. Gurina, Nadezhda A. Chalaya, Nadezhda M. Zoteyeva, Mariya A. Kuznetsova, Mariya P. Beketova, Oksana A. Muratova, Ekaterina A. Sokolova, Polina E. Drobyazina, and Emil E. Khavkin.
This manuscript is a resubmission of an earlier submission. The following is a list of the peer review reports and author responses from that submission.
Round 1
Reviewer 1 Report
The authors present a very interesting manuscript, but I can state that the form of processing is very problematic. Introduction - it is very extensive and actually after reading it, it is not necessary to continue reading the manuscript, because I have all the relevant information at my disposal and the authors do not really bring any new knowledge to their work. Before I get to the results and their discussion, I would like to comment on the methodology. here it is obvious that the authors recycle their work to some extent /especially reference no. 85 and the article by the author team in Agronomy/. At the same time, with the methodology, it is not clear whether the evaluation took place only for one year (especially the field one), which has a fundamental effect on the resistance evaluation (the influence of climatic conditions affects the occurrence of the pathogen, etc.). Information about climatic conditions is completely missing. The number of plants in the field collection is very different (2-10), which can significantly distort the achieved parameters. It is a pity that the authors did not at least identify the races of the pathogen in the given year, or at least biennial assessment. The presentation of the results and their discussion then suffer from methodological shortcomings, which, thanks to the form of the Introduction section, do not bring new knowledge in the given area. The Conclusion then actually summarizes and repeats the information at length. Even this part deserves to be reworked. It is clear from the above that I cannot recommend the publication of this manuscript.
Reviewer 2 Report
The manuscript is based on solid research and data. The methodology is appropriate and the results are significant. The study contributes substantially to the scientific knowledge and worth publishing. However, as any manuscript, it can be improved.
1. Introduction. Short information about potato importance in Russia, area, yield and challenges is well justified.
2. Table 1. The asterisk is not explained.
3. Tables 2 and 3. Heading – markers (the total number …). This perhaps means frequency of markers. Both tables are very busy. May be the number of total entries tested can be taken to separate column and range provided. In Table 2 for S. phureja last column gives frequency of 0.6 with 0 entries tested.
4. Lines 422-425: which year the test was conducted? Likewise line 413.
5. Tables S1. Would be nice to add a column with country of origin.
6. Conclusions look more like discussion. Can it be more focused on the main outcomes without references to literature and without references to tables and figures.
7. The main challenge of this paper is that the authors try to include so much material and results that the manuscript is difficult to follow and some important outcomes are probably hidden. In fact, this paper can be divided into three: introduction is a review paper with 94 references; distribution of the genes among the species is another paper; effect of genes on LB reaction is the third paper. However, if the authors like to keep it as such in one paper some simple conclusions are to be articulated: is there relationship between the laboratory and field tests; what are the key LB resistant species which can be utilized for practical breeding; what is the breeding implications/strategy to improve LB resistance based on the study.
Reviewer 3 Report
See attached file.
